# Proteolytic Cleavages in the VEGF Family: Generating Diversity among Angiogenic VEGFs, Essential for the Activation of Lymphangiogenic VEGFs

**DOI:** 10.3390/biology10020167

**Published:** 2021-02-23

**Authors:** Jaana Künnapuu, Honey Bokharaie, Michael Jeltsch

**Affiliations:** 1Drug Research Program, Faculty of Pharmacy, University of Helsinki, 00014 Helsinki, Finland; jaana.vulli@helsinki.fi (J.K.); honey.bokharaie@helsinki.fi (H.B.); 2Individualized Drug Therapy Research Program, Faculty of Medicine, University of Helsinki, 00014 Helsinki, Finland; 3Wihuri Research Institute, 00290 Helsinki, Finland

**Keywords:** vascular endothelial growth factors (VEGFs), VEGF-A, PlGF, VEGF-B, VEGF-C, VEGF-D, angiogenesis, lymphangiogenesis, CCBE1, proteases, ADAMTS3, plasmin, cathepsin D, KLK3, prostate-specific antigen (PSA), thrombin, wound healing, metastasis, proteolytic activation, vascular biology, lymphedema

## Abstract

**Simple Summary:**

Vascular endothelial growth factors (VEGFs) regulate the growth of blood and lymphatic vessels. Some of them induce the growth of blood vessels, and others the growth of lymphatic vessels. Blocking VEGF-A is used today to treat several types of cancer (“antiangiogenic therapy”). However, in other diseases, we would like to increase the activity of VEGFs. For example, VEGF-A could generate new blood vessels to protect from heart disease, and VEGF-C could generate new lymphatics to counteract lymphedema. Clinical trials are testing the latter concept at the moment. Because VEGF-C and VEGF-D are produced as inactive precursors, we propose that novel drugs could also target the enzymatic activation of VEGF-C and VEGF-D. However, because of the delicate balance between too much and too little vascular growth, a detailed understanding of the activation of the VEGFs is needed before such concepts can be converted into safe and efficacious therapies.

**Abstract:**

Specific proteolytic cleavages turn on, modify, or turn off the activity of vascular endothelial growth factors (VEGFs). Proteolysis is most prominent among the lymph­angiogenic VEGF-C and VEGF-D, which are synthesized as precursors that need to undergo enzymatic removal of their C- and N-terminal propeptides before they can activate their receptors. At least five different proteases mediate the activating cleavage of VEGF-C: plasmin, ADAMTS3, prostate-specific antigen, cathepsin D, and thrombin. All of these proteases except for ADAMTS3 can also activate VEGF-D. Processing by different proteases results in distinct forms of the “mature” growth factors, which differ in affinity and receptor activation potential. The “default” VEGF-C-activating enzyme ADAMTS3 does not activate VEGF-D, and therefore, VEGF-C and VEGF-D do function in different contexts. VEGF-C itself is also regulated in different contexts by distinct proteases. During embryonic development, ADAMTS3 activates VEGF-C. The other activating proteases are likely important for non-developmental lymphangiogenesis during, e.g., tissue regeneration, inflammation, immune response, and pathological tumor-associated lymphangiogenesis. The better we understand these events at the molecular level, the greater our chances of developing successful therapies targeting VEGF-C and VEGF-D for diseases involving the lymphatics such as lymphedema or cancer.

## 1. Introduction

In vertebrates, the family of vascular endothelial growth factors (VEGFs) typically comprises five genes: VEGF-A (in older literature often referred to simply as “VEGF”), placenta growth factor (PlGF), VEGF-B, VEGF-C, and VEGF-D. In addition to these orthodox VEGFs, several genes coding for VEGF-like molecules have been discovered in some members of the poxvirus and iridovirus families (collectively named VEGF-E) [1,2,3,4] and in venomous reptiles (collectively named VEGF-F) [5]. In vertebrates, the VEGF growth factors are central to the development and maintenance of the cardiovascular system and the lymphatic system. Non-vertebrates also feature VEGF-like molecules, but their functions are less well defined.

The subdivision of the vertebrate vascular system into the cardiovascular and the lymphatic system is reflected at the molecular level by a subdivision of the VEGF family into VEGFs acting primarily on blood vessels (VEGF-A, PlGF, and VEGF-B) and VEGFs acting mostly on lymphatic vessels (VEGF-C and VEGF-D). This specificity results from the expression pattern of the three VEGF receptors (VEGFRs). VEGFR-1 and VEGFR-2 are expressed on blood vascular endothelial cells (BECs), while lymphatic endothelial cells (LECs) express VEGFR-2 and VEGFR-3 (Figure 1).

The biology of the VEGFs and their signaling pathways has been extensively discussed elsewhere [10,11]. From all VEGF family members, only VEGF-A and VEGF-C are essential in the sense that constitutive ablation of their genes in mice results in embryonic lethality [12,13,14]. VEGF-A levels are so crucial that even heterozygous mice are not viable. In fact, *VEGFA* was the first gene where the deletion of a single allele was shown to be embryonically lethal [12,13]. While the primary function and importance of the cardiovascular system are also obvious to the layperson, the tasks of the lymphatic system escape even some life science professionals. Its major three tasks are:Tissue drainage for fluid balance and waste disposalImmune surveillance, including hosting and trafficking of immune cellsUptake of dietary long-chain fatty acids and other highly lipophilic compounds in the intestine

Considerable effort has been devoted to the mechanisms and effects of receptor binding and downstream signaling of the VEGFs. Less is known about the processes upstream of receptor binding such as secretion, release, and proteolytic processing. In this review, we want to briefly give an overview of what is known about the proteolytic processing of VEGFs with a focus on the lymphangiogenic VEGFs.

Evolutionarily, the importance of proteases has been remarkable. Proteolytic processing often regulates protein activity and creates variation in a protein’s function. This has been suggested by phylogenetic and functional studies in all kingdoms of life, including viruses [15], plants [16], and animals [17]. Not surprisingly, proteases are used to regulate function and create functional variety in the VEGF family and can be regarded as signaling molecules [18].

## 2. Proteolytic Processing of the Hemangiogenic VEGFs

Among the hemangiogenic VEGFs, protein diversification within a single VEGF family member relies more on differential mRNA splicing than on proteolytic processing (Figure 2; reviewed in [19]). mRNA splicing generates several isoforms of VEGF-A, which differ by the extent of the C-terminal, predominantly basic heparin-binding domain (HBD) [20,21,22]. The HBD mediates the interaction of VEGF-A with the extracellular matrix (ECM), cell surface heparan sulfate proteoglycans (HSPGs), and neuropilin-1. The interaction with HSPGs involves both a sequence-specific binding epitope and electrostatic effects of a predominantly basic amino acid sequence. Only a few isoforms are entirely devoid of heparin-binding properties under physiological conditions and therefore fully soluble. Mice expressing only the major soluble isoform (VEGF-A_121_) are born but show severe cardiovascular defects and die from cardiac failure [23].

The matrix-binding properties of the larger VEGF-A isoforms are essential for generating growth factor gradients, which are assumed to be essential for efficient organ vascularization [24,25]. VEGF-A_189_ and VEGF-A_206_ are sequestered in the extracellular matrix (or on cell surface HSPGs), and at least VEGF-A_189_ has been shown not to participate in receptor activation [26]. Proteases such as plasmin, urokinase-type plasminogen activator (uPA), and factor VII-activating protease (FSAP) can release and thus activate the ECM-bound, longer VEGF-A isoforms [21,27,28,29]. The cleavage of the primary isoform VEGF-A_165_ can also be mediated by various matrix metalloproteinases (MMPs), especially MMP-3, resulting in smaller, non-heparin-binding products [30]. While such cleavages do liberate VEGF-A and are necessary for the mitogenic activity of VEGF-A_189_ [26], they were reported to reduce the mitogenicity of VEGF-A_165_ [31]. Unfortunately, there is little insight into the nature of the molecular handover of HSPG- and ECM-bound VEGF-A to VEGFR-2 or the VEGFR-2/neuropilin signaling complex [11]. The isoform composition and the location where the cleavage happens are likely important determinants of the net effect. The release of cell surface HSPG-bound VEGF-A is perhaps more likely to result in productive signaling than the release of ECM-bound VEGF-A. Complementary to the ECM release by proteolytic cleavage of VEGF-A, enzymatic degradation of the ECM-binding sites, e.g., of HSPGs by heparinases, or binding site competition by heparin or heparan sulfate achieves the same release, but without loss of the HBD [27].

Of the four human PlGF isoforms, PlGF-2 and -4 also contain a C-terminal heparin-binding domain. At least the PlGF-2 HBD can be removed by plasmin [32]. VEGF-B_167_ also contains a heparin-binding domain homologous to the one in VEGF-A_165_, but it is unknown whether this domain is subject to proteolytic removal. A yet unknown protease unmasks the neuropilin-1 binding site of the longer VEGF-B_186_ isoform, but its target site is absent in VEGF-B_167_ [33]. The cleavage context suggests that thrombin can unmask the neuropilin-1 binding epitope (see Figure 2) [35]. Plasmin cleavage at the same site is likely but does not result in neuropilin-1 binding due to additional cleavages that remove important sequences for neuropilin-1 binding [33]. 

## 3. The Lymphangiogenic Growth Factors VEGF-C and VEGF-D

The hemangiogenic VEGFs are rendered inactive either through ECM-association or—as in the case for VEGF-A_189_—by their C-terminal auxiliary domain. Preventing receptor activation using inhibitory domains is also characteristic of the lymphangiogenic VEGFs. Upon secretion, VEGF-C and VEGF-D are kept inactive by their N- and C-terminal propeptides. Hence, the secreted forms are referred to as pro-VEGF-C and pro-VEGF-D. The removal of the propeptides requires two concerted proteolytic cleavages and happens in a very similar fashion for both VEGF-C and VEGF-D (see Figure 3):Protein convertases constitutively cleave VEGF-C before secretion. This intracellular cleavage occurs between the central VEGF homology domain (VHD) and the C-terminal propeptide. However, it does not remove the C-terminal propeptide because it remains covalently attached to the rest of the molecule by disulfide bonds [38,39,40].The second, extracellular cleavage activates the protein. This cleavage occurs between the N-terminal propeptide and the VHD [38] and can be mediated by different proteases. ADAMTS3 mediates VEGF-C activation in the embryonic development of the mammalian lymphatic system [41,42,43]. ADAMTS3 is specific for VEGF-C and does not activate VEGF-D. All other activating proteases target both VEGF-C and VEGF-D: plasmin [43,44], prostate-specific antigen (KLK3/PSA), cathepsin D (CatD) [45], and thrombin [46]. The resulting forms of VEGF-C and VEGF-D are referred to as active, mature, or short forms. However, they differ from each other at their N-termini because different proteases cleave at different positions within the linker between the N-terminal propeptide and the VHD (see Figure 4).

Interestingly, pro-VEGF-C can competitively block the receptor activation of active, mature VEGF-C. Its propeptides allow VEGF receptor binding but interfere with receptor activation. Apart from VEGFR-3, pro-VEGF-C also binds the co-receptor neuropilin-2. C-terminal propeptide processing exposes two terminal arginines (R226,227), which contribute to the conserved binding site for neuropilins [47]. Because it is not entirely clear whether pro-VEGF-C is completely incapable of receptor activation or whether it has some residual activity, pro-VEGF-C is either a partial agonist or an antagonist of mature VEGF-C [43].

## 4. Plasmin and Thrombin

The serine protease plasmin was the first protease that was shown to activate both VEGF-C and VEGF-D. Plasmin can remove both the N- and the C-terminal propeptides of VEGF-D to create a mature form containing only the VEGF homology domain [44]. One of plasmin’s main functions is to degrade fibrin, the main component of blood clots. Thrombin is the newest addition to the group of VEGF-C/D-activating enzymes. In addition to its classical role in converting soluble fibrinogen into insoluble fibrin fibrils during blood clotting, it plays a crucial role in early wound healing [46] by activating VEGF-C, which is released from ⍺-granules upon platelet aggregation [48]. Hence both thrombin and plasmin act concertedly to maintain a supply of active VEGF-C over the entire wound healing period.

However, in vitro, where no feedback loop exists to limit plasmin activity, prolonged exposure of VEGF-C results in VEGF-C inactivation [43]. In any case, without tissue damage, inactive prothrombin is not converted into thrombin, and inactive plasminogen not into plasmin. Therefore, in vivo, VEGF-C activation by thrombin or plasmin is likely restricted to situations with tissue damage. Using a similar rationale, platelet-rich plasma has been proposed for the treatment of lymphedema [49] and to promote wound healing [50]. Some proteomics studies have occasionally missed VEGF-C (as well as VEGF-A) when examining the platelet proteome, which might result from the relative resistance of the VEGF cystine knot to digestion with trypsin or similar proteases (unpublished data by the author). Nevertheless, other analyses and pharmacokinetic studies on anti-VEGF-C antibodies confirm the early findings of VEGF-C release during blood coagulation [51,52].

Plasmin activation of VEGF-C, which has been shown independently by two different groups [43,44], was not detected in a recent study [46]. Possibly, cleavage products might not have been recognized by the antibody due to low sensitivity or an absent epitope. Alternatively, the internal FLAG-tag preceding the cleavage site, which was used to prevent detection failure due to isoform-specific VEGF-C antibodies, might have interfered with the activation.

## 5. ADAMTS3 and the Cofactor CCBE1

ADAMTS3 was identified in the search for the endogenous protease that activates VEGF-C. Although plasmin had been identified as a VEGF-C-activating protease [44], it was never seriously considered as a physiological activator of VEGF-C due to its function in fibrin clot degradation. Moreover, lymphatic phenotypes have never been reported for the plasminogen knock-out mice [55] or human homozygous functional ablations [56].

In 2009, Alders used homozygosity mapping to identify mutations in the human *CCBE1* gene as a cause of Hennekam Syndrome (HS) [57], which is characterized by generalized lymphatic dysplasia [58]. When the same *Ccbe1* gene was ablated in zebrafish or mice [59,60], the phenotype was closely phenocopying the *Vegfc* knock-out [14]. Because it lacks any protease signature, CCBE1 was assumed to be somehow essential for the VEGF-C/VEGFR-3 signaling pathway, but not to be the VEGF-C-activating protease itself.

Co-transfection of CCBE1 with VEGF-C demonstrated that CCBE1 enhances the proteolytic processing of VEGF-C in 293T cells, and ADAMTS3 was identified as the responsible protease by mass spectrometric analysis of a partially purified CCBE1 from a CCBE1-overexpressing 293T cell line [43]. Based on in vitro data and its high homology to ADAMTS2, ADAMTS3 had been thought to function in the proteolytic maturation of procollagens [61]. In mice, *Adamts3* deletion does not lead to collagen fibril assembly deficiencies but instead aborts lymphatic development [42]. In humans, mutations in *ADAMTS3* have similarly been shown to result in a lymphatic phenotype, while symptoms associated with procollagen cleavage defects are absent [56]. Although these and other publications have confirmed that both ADAMTS3 and CCBE1 are required for successful pro-VEGF-C activation, a direct interaction between VEGF-C and CCBE1 has never been demonstrated [41,62,63].

Both domains of CCBE1 accelerate the activation of VEGF-C independently [43,54,58], using different mechanisms. While the N-terminal domain of CCBE1 appears to facilitate pro-VEGF-C encounters with ADAMTS3, the C-terminal domain acts like a coenzyme [64]. From all VEGF-C-activating enzymes, only ADAMTS3 and PSA/KLK3 have been shown to be influenced by CCBE1.

## 6. Species-Specific Differences

Based on sequence similarity, in vitro substrate, and domain organization, ADAMTS2, -3, and -14 form the aminoprocollagen peptidase subgroup within the ADAMTS protein family. Species-specific differences in the function of these proteases are seen in vertebrates. In zebrafish, Adamts3 and Adamts14 compensate for each other, and only the double *Adamts3/Adamts14* knock-out shows a lymphatic phenotype comparable to the *Vegfc* knock-out [65]. Such compensation does not happen in Adamts3-deficient mice, which are completely devoid of functional lymphatics [42]. Whether the observation that human ADAMTS14 can activate VEGF-C in vitro [65] reflects species differences among mammals or whether it is an observation without a physiological equivalent is still unknown.

Important species differences have also been reported for the growth factors. In mice, VEGF-D is dispensable for the development of the lymphatic system [53], while this is not the case in zebrafish, where it is, e.g., required to form the medial and lateral facial lymphatics [66,67]. However, even murine VEGF-D reportedly differs from human VEGF-D in its inability to interact with mouse VEGFR-2 [68]. Exactly the opposite seems to be the case in zebrafish, where the VEGF-D-VEGFR-3 interaction was reported to be absent [69], implying that lymphangiogenesis might happen in zebrafish independently of VEGFR-3. While direct demonstrations of the substrate specificities of the zebrafish Adamts proteases are still missing, it appears clear that zebrafish data are not easily extrapolated to mammals. Unfortunately, the same might be true for the extrapolation of mouse data to humans.

## 7. Which Cell Types Provide ADAMTS3 and CCBE1?

Since VEGF-C, ADAMTS3, and CCBE1 are all secreted proteins, immunohistochemistry cannot reveal their cellular origin. In the establishment of the early zebrafish lymphatics, *Pdgfra*-positive fibroblast populations appeared to be the source for *Vegfc*, *Adamts3*, *Adamts14,* and *Ccbe1*, as identified by single-cell RNA sequencing [65]. While in vitro data support the notion that fibroblasts are perhaps the dominating source for CCBE1 also in mammals [63], smooth muscle cells appear to make a significant contribution [14,70]. In some contexts, blood vascular endothelial cells appear to also be an important source of VEGF-C [71,72] and CCBE1 [73,74]. However, these are crude approximations of the actual cellular heterogeneity, and in non-homeostatic situations such as inflammation or cancer, other cell types, e.g., immune cells such as macrophages, are likely significant producers of both VEGF-C and VEGF-C-activating proteases [75,76,77].

## 8. Enigmatic Propeptides

The evolutionary origins of both propeptides of VEGF-C and VEGF-D are unclear. Unless assuming horizontal gene transfer, they have been conserved for hundreds of millions of years and can be found in virtually all invertebrate VEGF homologs [78,79,80]. Apart from the VEGFs, the only homologous sequences were found within larval silk proteins of the mosquito genus Chironomus [38,81], resulting in the nickname “silk homology domain” for the C-terminal propeptide.

Because disulfide bonds link the C- and the N-terminal propeptides of VEGF-C and VEGF-D, the first, constitutive cleavage by the protein convertase furin (or PC5 or PC7) does not remove any of the propeptides from VEGF-C or VEGF-D. Both propeptides are released simultaneously with the activating cleavage between the N-terminal propeptide and the VEGF homology domain (see Figure 3 and Figure 4). With 80 and 192 amino acid residues, respectively, the N- and C-terminal propeptides of VEGF-C are significantly longer than typical propeptides. They also fold independently and are therefore also often referred to as N- and C-terminal *domains*. According to the current understanding, the propeptides serve multiple functions.
The heparin-binding C-terminal propeptide mediates ECM-association and cell surface (HSPG) binding [63,82].Both propeptides collaborate in regulating receptor binding and activation [38,43]. While pro-VEGF-C binds VEGFR-3, it cannot (or only marginally) activate VEGFR-3. Thus, pro-VEGF-C and the individual VEGF-C propeptides, are competitive inhibitors of mature VEGF-C [43].The presence of the C-terminal propeptide is required for efficient cleavage of the N-terminal propeptide by ADAMTS3 [63].

Analogous to VEGF-A, the heparin-binding properties are likely necessary for the correct spatio-temporal distribution of the growth factor and its activity. When the VEGF-C propeptides are grafted upon VEGF-A, the resulting blood vasculature was denser compared with VEGF-A-induced vasculature [83]. Vice versa, when the C-terminal domain of VEGF-C was replaced by the heparin-binding domain of VEGF-A, less but larger lymphatic vessels were generated, which localized preferentially to HSPG-rich structures such as basement membranes [84]. The heparin binding of VEGF-C is somewhat weaker compared to that of VEGF-A. Although most heparin-binding affinity resides in the C-terminal propeptide, mature VEGF-C is a heparin-binding growth factor. VEGF-A_165_ binds tightest to heparin requiring 0.8 M NaCl for elution, while pro-VEGF-C and mature VEGF-C require elution concentrations of 0.435 and 0.265 M, respectively [82]. This might explain why both mature and pro-VEGF-C have a local effect and do not diffuse far [65]. That the C-terminal domain mediates the association or embedding of VEGF-C in the extracellular matrix was speculated shortly after its discovery [38] and was recently directly demonstrated in vitro [63]. Thus, pro-VEGF-C might be similar in this respect to latent TGF-β [85].

## 9. Changing Receptor Preferences with KLK3 and Cathepsin D

Based on N-terminal sequencing, two different mature, active forms were identified for VEGF-C and VEGF-D [38,86]. In the supernatant of 293 cells, the shorter mature form of VEGF-C was the dominant mature (“major”) form, while, for VEGF-D, the longer mature form was dominant. While this indicated early on that different proteases are involved in VEGF-C and VEGF-D activation, it remained unknown which proteases were involved. In 2011, Leppänen et al. found that the shorter (“minor”) form of active VEGF-D was not able to activate VEGFR-3 [87]. This finding was surprising as the activation of VEGFR-3 is considered a prerequisite of being lymphangiogenic. It also indicated for the first time that a lymphangiogenic growth factor can be converted into an angiogenic growth factor by proteolysis. At the same time, it explained why VEGF-D had been identified in some experimental settings as a powerful angiogenic growth factor [88]. While other research confirmed the disparity between VEGF-C and VEGF-D in terms of protease utilization for activation [41], the exact nature of the VEGF-D-activating proteases remained unknown until 2019 when Jha et al. tested whether their newly discovered VEGF-C-activating proteases PSA and Cathepsin D (CatD) could also activate VEGF-D [45]. In fact, CatD was able to generate the VEGFR-2-specific mature form of VEGF-D, which Leppänen et al. had described in 2011 [87]. Despite this, it remains to be shown which protease activates VEGF-D in vivo and whether there is a “physiological protease” equivalent to the VEGF-C-activating ADAMTS3. Perhaps VEGF-D is solely activated in non-homeostatic situations such as tissue damage. Nevertheless, also without any pathological challenge, VEGF-D knock-out mice display subtle alterations in some lymphatic networks [53,89]. These minor phenotypes could result from a lack of activated VEGF-D, but equally well from a lack of pro-VEGF-D (assuming it has some low level of activity) or possibly VEGF-C/VEGF-D heterodimers.

When comparing the effects of different VEGF-C- and VEGF-D-activating proteases [45], two trends are visible, which are summarized in Figure 5:The shorter the N-terminus of the resulting mature growth factor, the lower its receptor binding affinity and receptor activation potential.N-terminal shortening affects VEGF-C and VEGF-D very differently. While VEGF-C rapidly loses its potential to activate VEGFR-2 (through activation by ADAMTS3 or PSA), VEGF-D maintains much of its VEGFR-2 binding and activation potential. Vice versa, VEGF-D rapidly loses its VEGFR-3 binding and activation potential, whereas VEGF-C maintains much of it when processed to a similar degree.Both VEGF-C and VEGF-D are completely inactivated with respect to their receptor tyrosine kinase activity by complete removal of their N-terminal helices, which, e.g., can be achieved by prolonged exposure to plasmin.

## 10. Secondary Processing and Inactivation

At least in vitro, the longer forms of activated VEGF-C and VEGF-D can undergo additional cleavages, further shortening the N-terminus and modifying the receptor binding capabilities, e.g., CatD can remove the lymphangiogenic potential from plasmin-activated VEGF-D and the angiogenic potential from ADAMTS3-activated VEGF-C [45]. Finally, a cleavage by plasmin can inactivate VEGF-C and VEGF-D. However, such secondary (or tertiary) processing has not yet been demonstrated in vivo.

## 11. Other Cleavages

The activating, N-terminal cleavage of VEGF-D has also been proposed to be mediated by the protein convertases furin or PC5 [39]. While this is certainly a possibility, it seems unlikely that this represents a significant VEGF-D activation mechanism in vivo. The mature VEGF-D produced in furin-deficient Lovo cells upon transfection with furin could well be due to any other endogenous protease in Lovo cells. The requirement for furin in this system might occur if C-terminal furin processing was a prerequisite for the activating N-terminal cleavage. However, such a prerequisite seems not to exist for VEGF-C [41]. Vice versa, plasmin [44] or PSA [45] have not only been shown to perform the N-terminal processing of VEGF-D, but also the C-terminal processing. Nonetheless, this is also likely irrelevant since the protein convertases cleave constitutively inside the cell before VEGF-D ever has the chance to encounter plasmin or PSA.

## 12. Possible Involvement in Reproduction and Wound Healing

That VEGF-C is a possible substrate for kallikrein-like peptidases had been proposed before [90], but the identification of KLK3 (also known as prostate-specific antigen, PSA) was, nevertheless, surprising, especially because KLK3 is largely confined to sperm plasma. The presence of VEGF-C, CCBE1, and a VEGF-C-activating protease in sperm plasma [45] is too tempting not to speculate about a possible function of VEGF-C for reproductive biology, but such has not been confirmed yet. Mutations in KLK3 affect male fertility [91], but this is unsurprising since the main biological function of KLK3 is the degradation of gel-like seminogelins, which releases the sperm cells [92]. VEGF-A, which is also present in sperm plasma [93,94], had a modest effect on sperm motility [95]. Therefore, similar experiments were attempted with VEGF-C yielding very variable results (unpublished data by the author), perhaps due to the logistically challenging experimental setup.

Like KLK3, cathepsin D was also identified after an exhaustive analysis of bodily fluids for possible VEGF-C-cleaving activities [45]. VEGF-C is deposited by virtue of its C-terminal domain into the extracellular matrix and released by proteases [63]. Because VEGF-C accelerates wound healing [96,97], it appears possible that Cathepsin D provided by wound licking might activate latent ECM-embedded pro-VEGF-C. Thus, an instant angiogenic, lymphangiogenic, and immunologic stimulus would be provided. Compared to a single gene in humans, *KLK1* was several times duplicated in rodents, leading to at least 23 *KLK1* orthologs (some of which being pseudogenes), and some researchers believe that the evolutionary pressure to heal bite wounds rapidly and efficiently was driving this expansion [98].

## 13. Activating VEGF-C and VEGF-D in Cell Culture

While there are several cell lines that endogenously express VEGF-C or VEGF-D (most notably PC-3, from which VEGF-C was originally identified [99]), almost all experiments that require the expression of these growth factors have been performed by cDNA transfection. When the full-length wildtype cDNAs are used, the inactive pro-forms dominate in the cell culture supernatant of most cell lines (see Figure 3). Cells that express both CCBE1 and ADAMTS3 (such as cell lines derived from 293 cells) will process at least some of the pro-VEGF-C into mature, active VEGF-C. This endogenous background activation is sufficient to detect mature VEGF-C even in the absence of added proteases (see Figure 3). If these background activation bands are missing from a Western blot, the detection is likely not very sensitive, or something interferes with the physiological activation of VEGF-C by ADAMTS3. The degree of processing is relatively difficult to predict and appears to depend on cell density, stress level, cell culture medium, and—most importantly—VEGF-C expression levels. In any case, the processing is inefficient, and the 293T cell line that was used to generate the gel image in Figure 3 is among the cell lines that most efficiently activate VEGF-C endogenously.

## 14. Truncated cDNAs Are Used to Recombinantly Express Pre-Activated VEGF-C and VEGF-D

Therefore, when larger amounts of active VEGF-C are required, the solution has been to express a mutant VEGF-C cDNA, from which the sequences coding for the propeptides have been deleted (“ΔNΔC-VEGF-C”). All recombinant, commercially available VEGF-C and VEGF-D proteins are produced in this fashion. However, because the signal peptide’s cleavage context is disturbed, the N-terminus of the resulting protein can differ from the endogenously activated VEGF-C. Only N-terminal sequencing can reveal which form of VEGF-C is present. With few exceptions (R&D Systems), vendors do not provide this information. The same is true for many scientific publications that use truncated cDNAs to express VEGF-C or VEGF-D. While it is possible to predict the signal peptidase’s likely cleavage position, only N-terminal sequencing can give a definite answer. Many of the early experiments involving recombinant VEGF-D have used a truncated cDNA that results in a VEGF-D form, which is an intermediate between the VEGFR-2-monospecific and the VEGFR-2-/VEGFR-3-bispecific endogenous VEGF-D forms, making it difficult to interpret the data [68]. However, after the recent identification of the cleaving proteases, it became possible to generate specific mature forms by co-transfection of the protease with the full-length wildtype growth factor cDNA [43,45]. However, when using pre-activated forms of VEGF-C or VEGF-D, one should remember that it is unclear whether these exist as independent species in vivo. Pro-VEGF-C efficiently binds VEGFR-3 in the context of neuropilin-2, and the “in-situ” activation of pro-VEGF-C (while being bound to VEGFR-3) might be the standard mode of activation [43,63]. Interestingly, a transgenic mouse expressing pre-activated (ΔNΔC-VEGF-C) VEGF-C under the control of the keratin-14 promoter did not show the characteristic lymphatic phenotype in the skin as mice expressing VEGF-C from a full-length cDNA under the same promoter (unpublished data by the author) [100].

## 15. Modulation of Proteolytic Processing

Protease inhibitors have a veritable track record as drugs, targeting, e.g., viral proteases in AIDS and other viral infections [101], neutrophil elastase in lung diseases [102], and angiotensin-converting enzyme in cardiovascular diseases [103]. The opposite approach—promoting proteolysis—has also resulted in life-saving treatments, e.g., the use of tissue plasminogen activator (tPA) to dissolve blood clots in the immediate treatment of ischemic stroke [104].

Given the importance of the lymphatic system in many diseases [105], both VEGF-C and VEGF-D are likely worthwhile drug targets. Lymphedema, the swelling of organs or tissues due to an absent, hypoplastic, dysfunctional, or overloaded lymphatic network, represents a major clinical challenge because no causal, only symptomatic, treatments are available. The concept of pro-lymphangiogenic therapy to treat lymphedema has progressed to clinical trials using adenoviral VEGF-C gene therapy [106]. In these trials, pro-VEGF-C is produced from a full-length cDNA, relying on endogenous proteases for its activation. Using an adenovirus that produces pre-activated VEGF-C from a truncated cDNA would remove the requirement for endogenous proteases, which might or might not be a bottleneck. Alternatively, co-delivering CCBE1 and/or ADAMTS3 using an adenovirus cocktail could also boost the amount of active VEGF-C. However, the full-length cDNA of VEGF-C has been preferred over the truncated cDNA in most preclinical studies. Removing the N-terminal propeptide from VEGF-C results in an unpaired cysteine residue in VEGF-C’s receptor-binding domain. This extra cysteine residue is conserved among all VEGF-C and VEGF-D orthologs but is absent from all other VEGF family members [99,107,108]. Therefore, when active VEGF-C or VEGF-D are expressed directly from a truncated cDNA, the monomeric growth factor can be the predominant species [103]. Monomeric growth factor exposes the dimerization interface to the environment and is predicted to interfere with receptor dimerization and activation. For crystallization studies, the extra cysteine residues can be mutated to promote dimerization [8,82], but the use of mutated proteins as biological drugs requires solid justification.

A continuous low-level supply with VEGF-C appears necessary to maintain the structure and functionality of heavily engaged lymphatic networks [70,109,110]. As an alternative to VEGF-C itself, a highly specific VEGF-C-activating protease might equally be suitable if it can activate endogenous ECM-embedded VEGF-C. Such VEGF-C activation might both act via stimulating lymphatic pumping [111] and by inducing a compensatory expansion of the lymphatic network [112].

In inflammatory and infectious diseases, the lymphatic network must manage the fluid balance during inflammatory swelling. Perhaps more importantly, there is increasing evidence that both innate and adaptive immunological responses are crucially dependent on the lymphatics during all stages of an immune response [113,114]. Thus, the activation of VEGF-C could be used as a generic means to boost any immune response like an adjuvant.

## 16. Proteolytic Activation of VEGF-C and VEGF-D in Cancer

The crucial role of tumor-associated lymphatics for metastasis was recognized early on [115,116,117,118], and VEGF-C/VEGF-D inhibition has been proposed to therapeutically block metastasis. Since the tumor-promoting effects of VEGF-C and VEGF-D likely require proteolytic processing, inhibition could not only target the growth factors or receptors, but also the activating proteases. In vitro, VEGF-C-expressing MCF-7 and MDA-MB-435 cells, which have been used for xenograft tumor models, are inefficient in activating the growth factor [115,117]. Therefore, it is assumed that the activating proteases are supplied in these xenograft models by the stromal tumor compartment, perhaps by fibroblasts, inflammatory or endothelial cells [119,120]. Harris et al. generated a mutated form of VEGF-D, which is resistant to proteolytic activation They showed that this mutant could not promote tumor growth and lymph node metastasis in a mouse tumor model [121].

Tumor cells can migrate and form distant metastases via two distinct pathways: via blood vessels (hematogenic spread) and via lymphatics (lymphogenic spread). VEGF-C and VEGF-D can stimulate both pathways. By stimulating lymphangiogenesis into the tumor periphery (and occasionally also into the tumor), these growth factors maximize the access of tumor cells to the lymphatic vasculature. VEGF-C further appears to actively prepare the downstream lymph nodes for arriving cancer cells [122]. If the tumor happens to express suitable proteases, it is likely that VEGF-C (and even more so VEGF-D) is activated into forms that mimic VEGF-A but which are not inhibited by current anti-angiogenic treatments [123]. Such angiogenic redundancy might be one of the reasons why VEGF-A treatment is much less universal as initially anticipated, and why in amenable cancers, initial treatment success is usually followed by the development of resistance [124].

However, anti-lymphangiogenic therapy is a double-edged sword [125] because tumor-associated lymphatics are crucially important for the immune response against the tumor. When VEGF-C action was blocked in a mouse tumor model treated with immunotherapy, the mice receiving the anti-VEGF-C treatment died earlier than those who did not receive the treatment [126]. Similarly, in a mouse glioblastoma model, VEGF-C could amplify the CD8+ T cell response against the tumor [127]. In order to be able to successfully target VEGF-C in cancer, a thorough understanding of the underlying molecular mechanisms is needed. This understanding might ultimately allow us to separate the metastasis-enhancing function of VEGF-C from the immune-response-enhancing function of VEGF-C. Such separation might involve activating VEGF-C into a largely VEGFR-3-specific form eliminating some (but not all) of VEGF-C’s angiogenic features. However, since trafficking via the lymphatic neovasculature is integral to both tumor cell dissemination and immune response, separating these two functions appears unlikely to be feasible at the level of the VEGF-C/VEGFR-3 signaling axis. However, our understanding of the underlying molecular events is highly incomplete as we do not even know which specific proteases are activating VEGF-C in human cancers. A high-probability guess is that different proteases are involved depending on the cancer type.

## 17. Blocking VEGF-C and VEGF-D Activation

The proteolytic processing of VEGF-C and VEGF-D has been so far experimentally blocked only by mutagenesis of the cleavage sites. Furin and related protein convertases cleave VEGF-C after the double arginines (R226,227). Joukov et al. reported that mutating these arginines into serines (R226,227S) mostly blocked VEGF-C processing [38]. This is surprising since the N-terminal cleavage site should have been still subject to proteolytic attack because the first, constitutive C-terminal cleavage and the second, N-terminal cleavage were subsequently shown to occur independently of each other [41]. To generate an even more activation-resistant form of VEGF-D, Harris et al. mutated, in addition to the C-terminal cleavage site, the major N-terminal cleavage site and reported, similar to Joukov et al., almost complete abrogation of VEGF-D activation [121]. It is unclear why the unmutated minor N-terminal cleavage site in this protein did not result at least in partial activation. After all, in the same 293 EBNA cell line, the minor N-terminal cleavage site had been shown to account for approximately 20% of the activated protein [86]. However, a therapeutic effect might not require full inhibition of cleavage because pro-VEGF-C acts as an antagonist of mature VEGF-C, and therefore, a low level of cleavage might be acceptable [43].

## 18. Lymphedema and Genetic Lesions Affecting the Activation of VEGF-C

Lymphedema is traditionally categorized into either primary or secondary lymphedema. Most lymphedema cases fall into the secondary category, resulting from various external insults to the lymphatic system, surgery and infection being most common [128]. On the other hand, primary lymphedema results from genetic lesions, which can be inherited or acquired during development. The latter makes them more challenging to identify since the genetic lesion might be present only in a subset of cells or organs [129]. For roughly 40% of primary lymphedema cases, the underlying genetic lesion can be identified, and a clinical guide for this process has been established [130]. However, secondary lymphedema also has a genetic component [131,132,133], and thus, primary and secondary lymphedema should be considered the endpoints of a continuous spectrum.

All mutations that disrupt the VEGF-C/VEGFR-3 signaling pathway result in hereditary lymphedema (Figure 6). While the most common type of hereditary lymphedema is caused by a mutation in the VEGF-C receptor [134,135], any signaling pathway components can be affected, including the proteolytic activation of VEGF-C. Thus, in human lymphedema patients, disease-causing mutations have been found in VEGF-C itself [136,137], in its activating protease ADAMTS3 [62,63], and the cofactor CCBE1 [57,138]. Genetic lesions in *FAT4* result in a phenotype closely resembling the phenotypes caused by mutations in *CCBE1* or *ADAMTS3*. Only recently, Hennekam Syndrome was split into three different subtypes depending on the underlying genetic lesion (Hennekam Syndrome Type 1, 2, and 3).

Interestingly, the primary function of FAT4 is likely unrelated to VEGF-C processing or VEGFR-3 signaling. Instead, it appears necessary for the flow-dependent establishment of lymphatic endothelial cell polarity [139]. However, in vitro analysis of FAT4 has been hampered because it is a very large protein with a highly repetitive structure [140]. Besides VEGF-C/VEGFR-3 signaling pathway genes known to be compromised in hereditary lymphedema or Hennekam Syndrome, Table 1 also lists selected other genes known to be causative for hereditary diseases featuring lymphedema as a cardinal symptom. Furthermore, Table 1 contains, in addition to ADAMTS3, the genes of all other proteases reportedly able to activate VEGF-C and VEGF-D. For all of these, genetic lesions have been described, but lymphedema has not been associated with any of them, arguing that they are not required for the development of the lymphatic system.

## 19. Outlook: Molecular Nudging

With the first successes in Crispr-Cas clinical trials, genetic deficiencies within the VEGF-C/VEGFR-3 signaling pathways appear at least theoretically amenable for repair. However, even cutting-edge trials limit themselves at the moment to cells that can be easily modified ex vivo (blood diseases such as sickle cell disease and β-thalassemia) [152] or to very localized targets [153]. We are still far from a systemic repair of solid tissues, which would be needed since the lymphatic system penetrates almost all our bodies’ organs. Since at least a fraction of the VEGF-C appears to originate from blood vascular endothelial cells, a vascular-targeted repair appears possible [154]. If sufficiently specific, the systemic delivery of regulatory factors such as CCBE1 or ADAMTS3 might alternatively result in a widespread low-level activation (“molecular nudging”) of endogenous VEGF-C and a therapeutic effect. While such interventions do not reverse developmental routes already taken, they still might significantly improve life quality.

For cancer, being the prototype of a moving drug target, molecular nudging is not likely to have any impact. While a multitargeted anti-VEGF-A/-C/-D therapy might result in improved survival, any progress in this area will likely be incremental since using alternative tumor angiogenesis factors is only one of many escape mechanisms that tumors can deploy [124].

## Figures and Tables

**Figure 1 biology-10-00167-f001:**
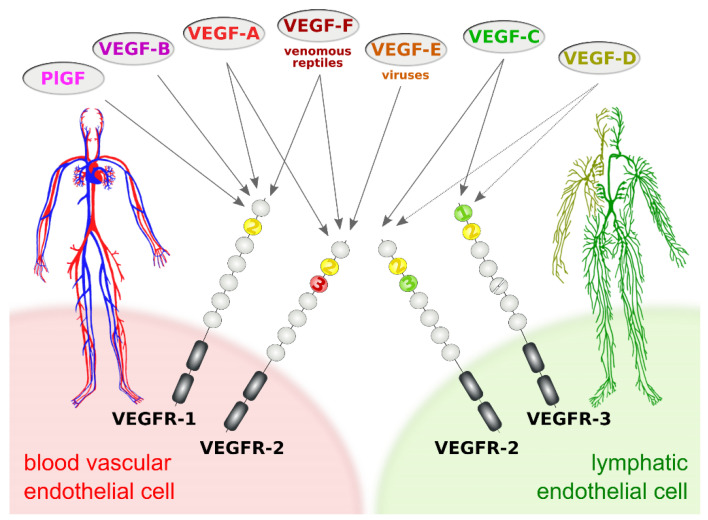
Vascular endothelial growth factors (VEGFs) act on blood vessels and/or lymphatic vessels depending on their affinities towards VEGF receptors 1, -2, and -3. VEGFR-2 is expressed on both blood and lymphatic endothelium. In principle, growth factors that do activate VEGFR-2 can promote both the growth of blood vessels (angiogenesis) and lymphatic vessels (lymphangiogenesis). VEGF-E and VEGF-F are not of human origin: VEGF-E genes are found in viral genomes, and VEGF-F is a snake venom component. All receptor-growth factor interactions require the extracellular domain 2 of the VEGF receptors (shown in yellow) [6,7,8,9]. Domain 3 of VEGFR-2 is important for the interaction of VEGFR-2 with both VEGF-A [7] and VEGF-C [8], and domain 1 of VEGFR-3 is important for the interaction of VEGF-C with VEGFR-3 [9].

**Figure 2 biology-10-00167-f002:**
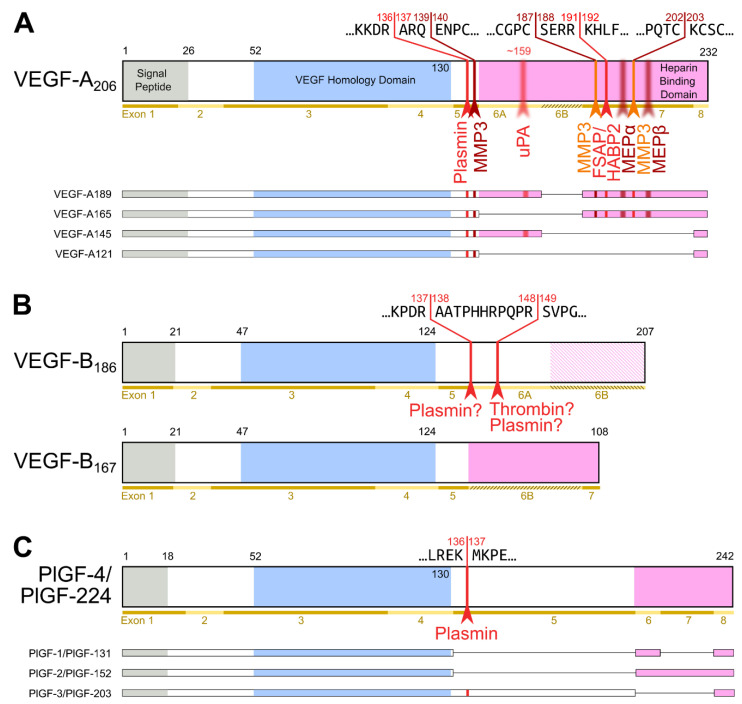
Most diversity among the hemangiogenic VEGFs is achieved by alternative splicing. Nevertheless, proteolytic processing of VEGF-A (**A**) [19] and placenta growth factor (PlGF) [32] (**C**) can convert the longer, heparin-binding isoforms into more soluble shorter species. (**B**) VEGF-B is a special case. Alternative splicing results in two isoforms that translate the same nucleotide sequence in two different frames resulting in a heparin-binding and a soluble isoform [33,34]. Due to the near-perfect cleavage context [35], thrombin has been suspected to be the responsible protease for VEGF-B_186_ cleavage [36]. Prothrombin is indeed expressed by 293T cells [37], in which the cleavage has been demonstrated [33]. Plasmin cleaves VEGF-B_186_ at at least four different sites, of which the two most likely predicted sites are indicated. Importantly, the predicted plasmin cleavage between Arg137 and Ala138 removes the interaction epitope for neuropilin-1 binding [33] Semi-transparent, blurry arrows indicate cleavages, for which only the approximate position is known. The figure shows only the most sensitive site from the plasmin cleavages of VEGF-A since prolonged incubation results in progressing degradation [30]. VEGF-B_186_ appears to be progressively degraded by plasmin as well [33]. For VEGF-A and PlGF, the numbering is according to the longest shown isoform. VEGF-A is cleaved not only by MMP3 but also in a similar fashion by MMP7, MMP9, MMP19, and - less efficiently - by MMP1 and MMP16 [30].

**Figure 3 biology-10-00167-f003:**
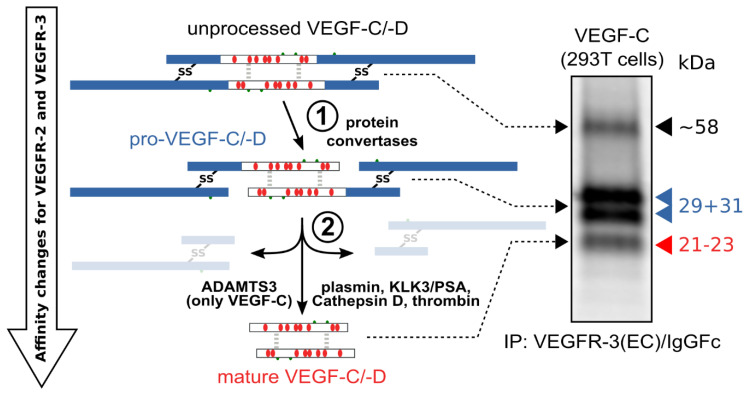
Two proteolytic cleavages are needed to activate VEGF-C and VEGF-D. The first cleavage, by protein convertases, is constitutive and intracellular. The second is highly regulated and happens after secretion of the pro-forms. Many different enzymes have been shown to catalyze the second cleavage, but the primary activating protease of VEGF-C in mammalian developmental lymphangiogenesis is A Disintegrin and Metalloprotease With Thrombospondin Motifs-3 (ADAMTS3). The immunoprecipitation (IP) of transfected 293T cells with a VEGFR-3(EC)/IgGFc fusion protein pulls down the 58 kDa full-length VEGF-C, the pro-VEGF-C peptides of 31 kDa and 29 kDa, and the mature VEGF-C. Proteins were resolved under reducing conditions by SDS-PAGE.

**Figure 4 biology-10-00167-f004:**
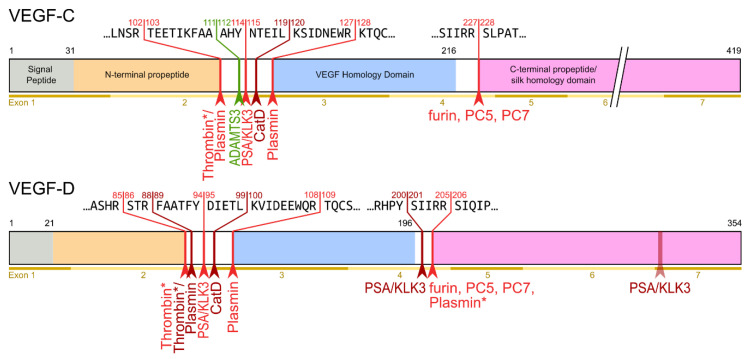
Human VEGF-C and -D are processed in a very similar fashion. The major difference between VEGF-C and -D is that ADAMTS3 activates VEGF-C, but not VEGF-D. This is one of the reasons why ADAMTS3 and VEGF-C are essential for lymphatic development and embryonic survival [14,42], whereas VEGF-D deletion in mice is well tolerated [53]. While the figure shows the exon structure of VEGF-C and -D, mRNA splice isoforms have only been reported for murine Vegfc [54]. The detected splice variants do not contain the full VEGF homology domain and are therefore not shown here. *Cleavage site is only predicted based on the amino acid context.

**Figure 5 biology-10-00167-f005:**
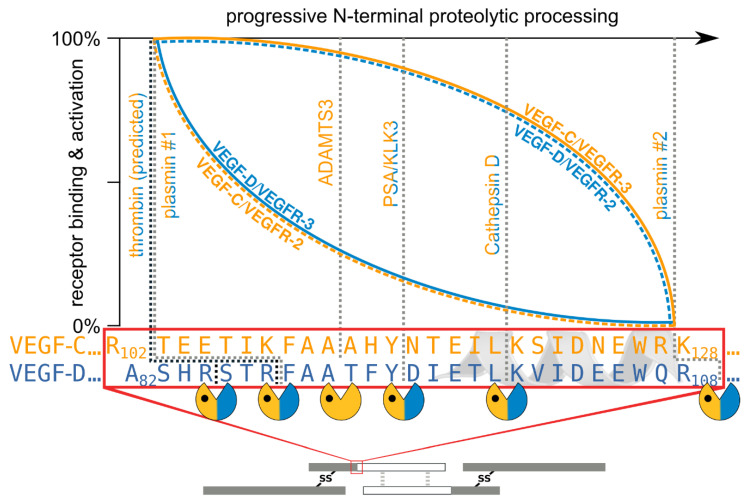
Biochemically, the group of lymphangiogenic activator enzymes is diverse. It includes a metalloproteinase (ADAMTS3), serine proteases (prostate-specific antigen (PSA)/KLK3, thrombin, and plasmin), and an aspartic protease (cathepsin D). VEGF-C and VEGF-D share all activating enzymes except for the most important one: ADAMTS3. ADAMTS3 is exclusive for VEGF-C and required for the physiologic activation of VEGF-C during developmental lymphangiogenesis [41,42,43]. VEGF-C and VEGF-D are differently affected by proteolytic processing. With progressing processing, VEGF-C largely maintains its lymphangiogenic properties but loses its angiogenic properties quickly. VEGF-D behaves precisely the opposite way: processing with Cathepsin D almost completely abolishes its lymphangiogenic properties and fully unmasks its angiogenic properties [45]. Extensive exposure of both VEGF-C and VEGF-D to plasmin abolishes all VEGFR-2 and VEGFR-3 binding properties.

**Figure 6 biology-10-00167-f006:**
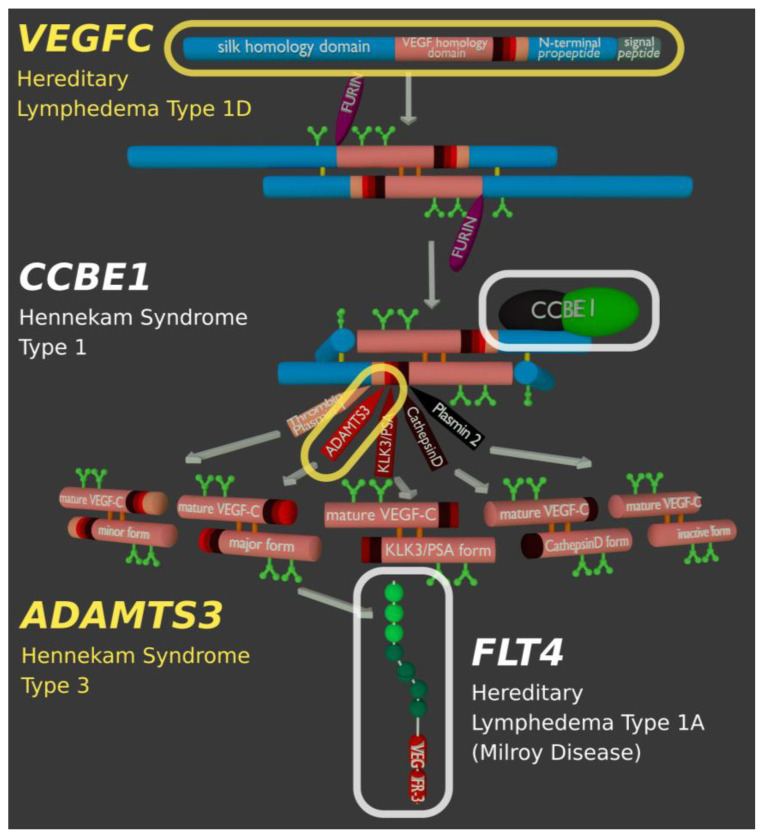
Genes of the VEGF-C/VEGFR-3 signaling pathway known to be involved in hereditary lymphedema. Although different components of the signaling pathway can be affected, the largest fraction of cases are caused by mutations in the *FLT4* gene, which codes for the VEGF-C receptor VEGFR-3.

**Table 1 biology-10-00167-t001:** Important genes for lymphatic development and/or VEGF-C activation, which are involved in human hereditary disorders. Mutations in the VEGF-C-activating proteases are not associated with any lymphatic phenotype except for ADAMTS3, arguing that only ADAMTS3 is essential for lymphatic development.

Gene	Protein	Human Disease (OMIM)	Remarks
Genes within the VEGF-C/VEGFR-3 signaling pathway; for clinical details see [130], and for molecular details [141]			
*VEGFC*	Vascular Endothelial Growth Factor-C (VEGF-C)	Hereditary lymphedema type 1D (615907)	VEGF-C is the primary growth factor for lymphatic endothelial cells.
*FLT4*	Vascular Endothelial Growth Factor Receptor-3 (VEGFR-3)	Hereditary lymphedema type 1A (Milroy disease, 153100)	VEGFR-3 is the primary receptor of VEGF-C.
*CCBE1*	Collagen and calcium-binding EGF domain-containing protein 1	Hennekam lymphangiectasia-lymphedema syndrome type 1 (235510)	Enhances the processing of VEGF-C by ADAMTS3 and KLK3.
*ADAMTS3*	A disintegrin and metalloproteinase with thrombospondin motifs 3	Hennekam lymphangiectasia-lymphedema syndrome type 3 (618154)	ADAMTS3 catalyzes the final step in the activation of VEGF-C.
Genes coding for proteases that can activate VEGF-C and/or VEGF-D (no lymphatic phenotype reported)			
*ADAMTS14*	A disintegrin and metalloproteinase with thrombospondin motifs 14	Association with age of onset in tendinopathy [142]	ADAMTS14 can activate VEGF-C in vitro [65].
*KLK3*	Kallikrein-like peptidase 3, Prostate-specific antigen (PSA)	Association with human fertility [91]	KLK3/PSA is most commonly known as a prostate cancer marker [143].
*CTSD*	Cathepsin D	Neuronal Ceroid Lipofuscinosis type 10 (610127)	CTSD deficiency causes a neurodegenerative disorder [144].
*F2*	Prothrombin/Thrombin	Hereditary thrombophilia type 1 (188050)	Specific F2 mutations increase the risk of venous thromboembolism [145]. Thrombin potentiates vascular endothelial growth factor- (VEGF-) induced endothelial cell proliferation [146].
*PLG*	Plasminogen/Plasmin	Plasminogen deficiency type 1 (217090)	PLG deficiency leads to pathological fibrin deposition but no increased risk of thrombosis [147].
Selected other “lymphedema” genes; for a comprehensive listing, see [130]			
*FOXC2*	Forkhead box protein C2	Lymphedema distichiasis syndrome (153400)	The maturation of lymphatic vessels and the formation of lymphatic valves requires FOXC2 [148,149].
*GJC2*	Connexin 47	Hereditary lymphedema type 1C (613480)	CJC2 is a gap junction protein that enables communication between lymphatic endothelial cells [150].
*FAT4*	Protocadherin Fat 4	Hennekam lymphangiectasia-lymphedema syndrome type 2 (616006)	FAT4 is required for lymphatic endothelial cell polarity and might influence VEGFR-3 signaling [139].
		Van Maldergem syndrome type 2 (615546)	Van Maldergem syndrome 2 has overlapping features with Hennekam syndrome type 2 but none or only infrequent lymphatic involvement [151].

## Data Availability

Not applicable.

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
