# Peer review of "Proteolytic Cleavages in the VEGF Family: Generating Diversity among Angiogenic VEGFs, Essential for the Activation of Lymphangiogenic VEGFs"

_biology, 2021, doi:10.3390/biology10020167_

Round 1

Reviewer 1 Report

It was a great pleasure to read the review article entitled ‘The proteolytic activation of vascular endothelial growth factors’ by Künnapuu et al. The review addresses the important and cutting-edge topic of vascular endothelial growth factor biology in development and disease, with a main focus on the lymph-angiogenic growth factors. It summarizes the biological functions and mechanisms of activation of vascular endothelial growth factors and their current / potential future application in human disease.

Strength: The review is well written and  provides a great overview on the proteolytic activation of VEGFs. In addition, it highlights several interesting findings on VEGF biology which are not common knowledge and which can be inspiring for new studies (e.g. expression of VEGF-C in sperms). As lymphovascular medicine/biology is not only limited to the skin or vasculature but the entire body, it provides new impulses for transdisciplinary investigation on the role of the lymphatic system and lymphatic growth factors.

The figures are well organized and convincing.

Weaknesses: Although the review is well written and provides a very detailed  overview on VEGFs, the paragraph on VEGFs and hereditary diseases is quite short. Especially as authors discuss the role of VEGFs in human disease, a more detailed paragraph on primary lymphatic diseases (lymphedema) is required. It would be very interesting to see a table with information about genes and phenotype associated with vascular endothelial growth factor activation and growth factors and receptors itselves. It would be very helpful if also the name of the diseases is included (e.g. Milroy disease). This could be very important to broaden the spectrum of readers, also from the medical and genetics field. In addition, I would like to ask the authors to comment on the role of FAT4 in VEGF-C processing as mutations in FAT4 cause Hennekam lymphangiectasia-lymphedema syndrome 2 (similar phenotype to CCBE1 mutations).

Additionally, I have the following minor comments:

Line 36/37: The authors introduce VEGF-F earlier than than VEGF-E. Please arrange it in an alphabetic order. 

line 64: As fatty acids are predominantly been uptaken in the intestine, I recommend using intestine instead of gut.

line 174-176: 'Plasmin is able to remove both the N- and the C-terminal propeptides of VEGF-D to create a mature form containing only the VEGF homology domain'

Please include a short comment on mechanism of VEGF-C activation by plasmin. 

201-205: 'At least in mice, Adamts3 deletion does not lead to deficiencies in collagen fibril assembly but instead aborts lymphatic development [38]. Subsequent publications confirmed that both ADAMTS3 and CCBE1 are required for successful pro-VEGF-C activation but interestingly, a direct interaction between VEGF-C and CCBE1 has never been demonstrated.'

Based on this statement it is not clear whether there is evidence on Adamts3-mediated VEGF-C activation only in mice or not. As you are already citing Brouillard et al., it would be helpful for the reader to make clear that ADAMTS3 is also required in humans. 

Author Response

Thank you for your constructive comments. Please find our point-to-point answers in the attached PDF file!

Reviewer 2 Report

The review by Künnapuu, Bokharaie and Jeltsch gives an overview of the proteolytic activation of the lymphangiogenic factors VEGFs, with a focus on VEGF-C and VEGF-D.

It is a very interesting review. And the information presented appears accurate, up to date and relevant. However, I feel that the review could be improved in term of style.

For example, figure 1 could  use multiple panels .

Style could be simplified to ease reading. I think this article will be of interest for a broad international audience. So, it might be worse simplifying the writing style.

Sometimes the style is indirect, uses long sentences and a bit verbose.

Abstract: While the abstract presents nicely the function of proteases in different context, I think it could highlight better the benefit of studying the proteolytic regulation of VEGF-C and VEGF-D. It might just involve modifying the flow of ideas to better highlight the importance of  these cleavages from a biomedical point of view or an evolutionary point of view and the importance of lymphangiogenesis. 

Typo: Authors wrote lymphangiogenesis (page line 21) and then lymphangio-genesis (line 22). Please ensure consistency.

Introduction:

Style.

Lines 50 to 52: Figure 1: The legend could be simplified. The following sentence seems long: “Growth factors that are able to activate VEGFR-2 can in principle promote both the growth of blood vessels (angiogenesis) and lymphatic vessels (lymphangiogenesis) since VEGFR-2 is expressed on both blood and lymphatic endothelium.”

Figure 2: The legend is long. It would be valuable to add some panels and break down the legend according to the panels. For example: A for VEGF-A, B for VEGF-B, C for PIGF. 

Line 57 to 59: I wonder if the flow of idea could be improved for the two sentences that ends with ‘’in embryonic lethality “. While the two sentences present different information; at first, they seem redundant.

Line 70: Should a title or subheading be added here?

Lines 93 to 102: It might be valuable for the authors to add reference to figure within this paragraph.

Line136 to 143: The figure 3 is not ideally position in the text and impairs the flow of reading. I would suggest moving it to another location maybe line 128.

Line 161 to 163: Please simplify figure legends and improve flow of idea. The following sentences are fragmented and somewhat difficult to read. ‘However, ADAMTS3, which is essential for lymphatic development, cannot activate VEGF-D (at least not in mammals). mRNA splice isoforms of VEGF-C have only been reported in mice [45], but these do not contain the full VEGF homology domain and are therefore not shown here.’ 

Is ADAMTS3 important to lymphatic development because it cannot activate VEGF-D or because it cleaves VEGF-C? Or because this is the only protease cleaving VEGF-C only?

167: avoid double negation.

183, Is a reference missing here?

187-190: Please create two sentences here. 

191-193: style. ‘In 2009, Alders et al. had shown by homozygosity mapping, that mutations in the human CCBE1 gene can cause Hennekam Syndrome, which is characterized by generalized lymphatic dysplasia [48].’ Please see suggestion of editing: ‘In 2009, Alders used homozygosity mapping and identified mutations in the human …’’

197 not sure ‘was able’ is required in the sentence ‘could enhance’ or ‘enhances’.

199: ‘mass spec analysis’, I suggest to use the whole word ‘spectrometry’

213: I am unsure about what is meant by 'compensate'?

Have redundant function as adamts3 ko or adamts14 ko retain the capacity to cleaved VEGF-C? 

And in mice when is meant by the absence of compensation? Now I am curious and want to know more. Is it based on morphological or functional changes observed in the lymphatic vessels during development?

217, ‘Also on the growth factor side’ .. seems an awkward transition here.

219: 'partial compensation'… what is meant there? Is it based on the morphology of the lymphatic system?

267: ‘also’ is not necessary in this sentence.

268: It might be valuable to explain why NaCl elution concentrations are reported here.

288-289. Is ‘despite this’ truly necessary in this sentence.

300: suggestion of edits: ‘Despite this similarity, the shortening affects VEGF-C and VEGF-D very differently. ‘ ‘This shortening affects VEGF-C…’

347” Is a reference needed regarding expression of VEGF-C  and CCBE1 in sperm plasma? Or are the authors already referring to the unpublished work.

356- 357: Authors might want to generate two sentences.

372: Are the authors referring to western blotting conditions that are SDS page without beta-mercaptoethanol? Or even with beta-mercaptoehtanol? native versus denaturating?

407 ‘Given the importance of the lymphatic system in many diseases [95] both VEGF-C and VEGF-D are likely worthwhile drug targets.’ I wonder if some of the notions presented in this sentence could be presented at the beginning of the abstract. With a focus on the proteolytic regulation of VEGF-C and VEGF-D. This will highlight immediately the importance of this field of research.

464-465, editing might be needed as it is confusing for the reader to understand what is truly meant by ‘activation block’. Is it just that there is no need for a full inhibition?

Author Response

(The authors gave the same response as above.)

Reviewer 3 Report

The review focuses on the lyphamgiogenic factors VEGF-C and D and the specific proteolytic cleavages they undergo for activation. The paper is well structured and try to explain the different behaviours of the factors in different contexts and tissues where the proteolytic enzymes are variabley expressed or active. The open or debated points are discussed as the role of lymangiogenesis in tumors, providing clues to the actuality of the theme.

The review is well written and chomprehensive. I only suggest to check the text at lines 312-313and 322-323 since the sentences are not complete,

Author Response

Thank you for your encouraging comments! Both incomplete sentences are now fixed.

Reviewer 4 Report

The article review by J. Künnapuu et al., investigate the proteolytic factors involved in the activation of vascular endothelial growth factors. The authors have mainly focused their study on proteolysis processes among the lymphangiogenic VEGF-C and VEGF-D.

The article is interesting and well constructed and argued, with a lot of emphasis on the lymphangiogenic growth factors VEGF-C and VEGF-D.

However, the family of VEGFs comprises five factors that are: VEGF-A, PlGF, VEGF-B, VEGF-C, and VEGF-D.

Therefore, the major key points to be improved are:

1- to better define the impact of current therapies that may be of strategic interest on the target of activating protease factors VEGF-C and VEGF-D.

2- to describe in more details the current therapeutic approaches in mice and/or humans that regulate mechanisms able to inhibit the metastasis-enhancing from the immune response-enhancing function of VEGF-C and/or VEGF-D. Or to illustrate hypotheses on it.

3- Perhaps the title should be rewritten to better explain the whole study on proteases and lymphangiogenesis.

4- In Figure 1, authors should explain better why the VEGFR-2 receptor has a red dot and a yellow dot? This from the text is not clear to the reader.

Author Response

Thank you for your encouraging and helpful comments! Please find our point-to-point responses in the attached PDF file.

Round 2

Reviewer 4 Report

The authors in this new version have responded to all the criticisms raised in a satisfactory way. The article is well written with fluent English.

There are no other requests for changes.